# Wearable Sensor Clothing for Body Movement Measurement during Physical Activities in Healthcare

**DOI:** 10.3390/s21062068

**Published:** 2021-03-16

**Authors:** Armands Ancans, Modris Greitans, Ricards Cacurs, Beate Banga, Artis Rozentals

**Affiliations:** Institute of Electronics and Computer Science, 14 Dzerbenes St., LV-1006 Riga, Latvia; modris_greitans@edi.lv (M.G.); ricards.cacurs@edi.lv (R.C.); beate.banga@edi.lv (B.B.); artis.rozentals@edi.lv (A.R.)

**Keywords:** human movement analysis, wearable sensor network, 3D motion capture, inertial sensors, kinematics, body area networks

## Abstract

This paper presents a wearable wireless system for measuring human body activities, consisting of small inertial sensor nodes and the main hub for data transmission via Bluetooth for further analysis. Unlike optical and ultrasonic technologies, the proposed solution has no movement restrictions, such as the requirement to stay in the line of sight, and it provides information on the dynamics of the human body’s poses regardless of its location. The problem of the correct placement of sensors on the body is considered, a simplified architecture of the wearable clothing is described, an experimental set-up is developed and tests are performed. The system has been tested by performing several physical exercises and comparing the performance with the commercially available BTS Bioengineering SMART DX motion capture system. The results show that our solution is more suitable for complex exercises as the system based on digital cameras tends to lose some markers. The proposed wearable sensor clothing can be used as a multi-purpose data acquisition device for application-specific data analysis, thus providing an automated tool for scientists and doctors to measure patient’s body movements.

## 1. Introduction

The development of body movement measuring systems is essential in telemedicine and the provision of remote medical services, which are topical issues today. A striking example is an outbreak of the new COVID-19 virus, which has forced people to stay at home, where they often do not have access to a doctor. With 3-dimensional (3D) body motion capture, it is possible to draw more complete consultations in areas such as physiotherapy, rehabilitation, sports and elsewhere.

There are various kinds of motion-sensing technologies. Firstly, optical sensor systems should be noted. They can consist of optical markers (active or passive), placed on the human body. They use visual-sensing cameras providing high accuracy, but the downside for these kinds of sensors is their limited freedom of movement, due to the use of stationary cameras and occlusion [1].

As well as optical sensors, there are ultrasonic sensors. Unlike optical sensors, ultrasonic sensors operate at a close distance, up to 17 m in the line of sight, with state-of-the-art technology [2].

Furthermore, Metamotion (Metamotion Ltd, London, UK) offers motion-tracking using an exoskeleton. Compared with other-motion sensing technologies, its exoskeleton is significantly heavier, and it is harder to hide its visual form. As a result, exoskeletons are not comfortable to use.

Electromagnetic sensors are also used for movement tracking. They acquire high accuracy (sub-millimeter and sub-degree) with no motion constraints, such as line of sight, which is required by optical and ultrasound sensing technologies [1]. However, the accuracy of electromagnetic sensors does degrade with increasing distance between the electromagnetic field generator and sensor locations. The accuracy of this technology also tends to alter with the changing working environment and orientation of sensors [1].

Last, but not least, there are inertial measurement units (IMUs), which are not limited to motion constraints, like line of sight, or movement restrictions, like room or lighting conditions, as is the case for camera systems. An overview of motion capture technologies is given in Table 1.

IMU-based motion-tracking systems can facilitate the use of 50+ sensors simultaneously (Nansense suit from Nansense Inc, Los Angeles, CA, USA), and, for most applications, are sufficient for body-limb-pose and hands-gesture estimation. Considering the use of body-movement measurements in medical applications, especially in the case of telemedicine, IMU-based solutions are potentially the most appropriate technology for the development of a low-cost, easy-to-use system. In the global market, several IMU-based motion-capture products, such as Xsens (Xsens Technologies B.V., Enschede, The Netherlands), Shadow Motion (Motion Workshop, Seatle, WA, USA), STT Systems (STT INGENIERÍA Y SISTEMAS, SL, San Sebastián, Spain), Nansense (nansense Inc, Los Angeles, CA, USA), Rokoko (Rokoko Electronics, Copenhagen, Denmark), Perception Neuron (Noitom Ltd, Miami, Trivisio, (TRIVISIO Prototyping GmbH, Trier, Germany), Florida), AiQ-Synertia(AiQ-Synertia Ltd, Tampei, Taiwan), are available. The overview of IMU motion-capture technologies mentioned above is given in Table 2.

Several studies comparing IMU-based motion-capture systems with optical systems for different regions of the body have been carried out in the literature [3,4,5,6,7,8,9,10]. The focus of our research is a universal, whole-body motion-capture system using wearable IMU sensors. Based on the literature mentioned above, the following criteria of movements for testing such systems are selected:Exercise duration is more than 1 min;Exercises exploit as many joint degrees of freedom as possible;The movements are made with the highest possible amplitude.

The study aims to offer an efficient hardware and middleware solution for IMU-based sensor clothing, intended as a multi-purpose data acquisition device for application-specific data analysis. For example, this solution could be used as an automated tool for scientists and doctors to measure patient body movements. The IMU system performance summarized in Table 2 defines the system requirements for this study, but its cost is significantly reduced. This article suggests a simplified architecture, more convenient sensor connection embedded in the suit and the reconstruction of universally usable data of body pose.

Figure 1 shows a step-by-step overview of the research conducted in this paper. Accordingly, the rest of the paper is organized as follows: the next section discusses the correct placement of sensors on the human body, as this is critical for obtaining correct data. Section 3 introduces the architecture of the proposed system, followed by a study of sensor system implementation in clothing to create a safe and user-friendly solution. Section 5 is devoted to the reconstruction of the human skeleton 3D model and movement. The developed experimental set-up and obtained results are presented in Section 6, and in the conclusions, further possible research directions and discussions are summarized.

## 2. Sensor Placement

The human body is made up of an average of 206 bones, and forms a complex mechanical system. For sensor-node-placement selection, it is convenient to divide the body into zones and perform motion assessment individually to decide which movement is involved in which zone. Of course, since the human body is a unified whole, all movements naturally involve all areas of the body. However, in this case, only areas recognized by the medical staff and patients as making up the movement during a certain exercise or procedure are considered.

By examining the various movements and evaluating which part of the body is involved in these movements, the following main areas of focus were selected: neck, spine, shoulder, elbow, hip, knee, ankle and wrist [11].

To correctly measure the movements of selected areas, it is necessary to choose measurement points accurately. Based on body-part movement directions, there are three types of joints—uniaxial, biaxial and multi-axial. Each type of joint has dedicated functions ensuring the mobility and stability of the body, which is an integral part of quality of life [12,13]. Sensor placement must be chosen so that it is possible to measure key points for human motion analysis while taking human biomechanics and anatomic structure into account. Knowing the biomechanics of measured joints allows for the minimization of the sensor count, a reduction in power consumption and the design of a less cumbersome sensor network for the user.

Joint movement is created by muscle contractions. Muscles connect two body parts, and when they change length, the angle between body parts is changed. When muscle contraction takes place, muscle cells called myofibrils shorten. This causes the whole muscle to become shorter [14]. As the muscle becomes shorter, it widens as well, causing errors in sensor readings.

Figure 2 shows the sensor placement on biceps brachii during the elbow flexion and extension. When in the flexed state, the biceps brachii muscle is shortened and widened in the center. We can see that, although the orientation of the upper arm does not change, the sensor orientation changes (green box with red arrow), introducing a measurement error into the orientation of the upper arm. Therefore, sensor placement on body parts with minimal muscle tissue can reduce the errors caused by muscle contractions. For ligament flexion or muscle contraction in a particular body part, sensors must be chosen to minimize surface relief changes.

Taking previous considerations into account, the sensor placement seen in Figure 3 is proposed. Spine movements are measured with sensor 1, which is attached to the back of the head, and spine sensors 2 to 4. Left and right shoulder movements are measured with sensors 5 and 6, attached on the top surface of the shoulder, while elbow movements are measured with upper arm sensors 7 and 8 and lower arm sensors 9 and 10. Hip movement is measured with spine sensor 4 and upper leg sensors 11 and 12. Left and right knee movement is measured with upper leg sensors 11 and 12 and lower leg sensors 13 and 14. Left and right ankle movements are measured with lower leg sensors 13 and 14 and feet sensors 15 and 16. Additional sensors 17 and 18 may be used to measure wrist movement by measuring the angle between lower arm sensors 9 and 10 and hand sensors 17 and 18.

## 3. Sensor Network Architecture

To synchronously collect IMU sensor data from the human body locations specified above, network architectures of IMU-based motion-tracking suits in Table 2 were analyzed. According to this table, architectures of IMU motion-tracking suits can be classified into two main categories: wired and wireless. One of the main advantages of wearable systems using wireless sensors is the convent data transmission over the air (Xsens Awinda, Perception Neuron Pro, AiQ-Synertial IGS Cobra Suit, STT Systems iSen). In this way, the major challenge for wire integration into the clothing for wired networks is obsolete. However, wired IMU motion-tracking systems are still widely exploited (Xsesns Link, Shadow motion capture system, Nansense INDIE full-body motion capture suit, Rokoko Smartsuit Pro, AiQ-Synertial wired version of IGS Cobra Suit) because of their higher data rates, easier power management and more reliable communication channels.

The available information about the implementation of IMU motion-tracking suits discussed above is very limited. However, based on the technical information published in the official web pages, they all facilitate multiple branches with a series of sensors connected to a central hub in parallel for data acquisition and transmission to an auxiliary device for further processing. The advantage of this sensor network configuration is optimized wiring, as each branch connects sensors from specific areas of the body.

Often, a multi-branch solution is based on a bus topology that allows each sensor to be addressed, as well as its data read [15]. Our approach is to take into account that the data are collected in a certain order from all sensors, which allows the sensors and their data to be identified by their location in the network topology. In this way, it is possible to simplify communication between the sensors, because, unlike the bus approach, there is no need to address the sensors.

Based on the considerations above, the architecture of a sensor network in Figure 4 is proposed. It facilitates synchronous real-time sensor data acquisition from multiple sensor branches connecting sensors from different body regions (arms, legs, back, neck, etc.). In each branch, sensor nodes are connected in enhanced serial peripheral interface (SPI) daisy-chain configuration, supporting baud rates up to several megahertz without data overhead for device addressing. It provides relatively fast data transfer speeds (200 sensor nodes at 50 Hz), which are feasible for real-time applications [16].

The multi-branch SPI daisy-chain configuration allows for synchronized real-time sampling design and centralized power supply for sensor nodes with only four wired connections. Compared to wireless channels, wired connections are less susceptible to environmental interference sources, especially in the crowded 2.4 GHz ISM band [17]. Wired connections embedded in the clothing can provide power delivery using one centralized power source. This simplifies charging and eliminates the need for separate power sources for each node.

A detailed architecture of the sensor node is shown in Figure 5. Each sensor node accommodates a single inertial measurement unit (IMU), comprising an accelerometer, gyroscope and magnetometer sensors. Advanced IMUs may have additionally embedded algorithms for sensor data preprocessing, calibration and fusion into orientation data. The microcontroller (MCU) is used for synchronized sensor data acquisition through an I2C interface and transmission through an SPI daisy chain configuration to the central data acquisition node.

The architecture of the central node is shown in Figure 4. It has a centralized power management for all sensor nodes using a single rechargeable battery. The power management includes a switch-mode power supply for the efficient stabilization of system voltage, a battery-charging circuit for battery charging from a computer or a wall adapter and a battery-level measurement circuit for battery-level monitoring. In the current scenario, the wire resistance is considered negligible, and the supply voltage is regulated only on the master node. However, in some scenarios, an excessive voltage drop on power connections can disturb reliable operation of the sensor network. In those cases, additional power management on sensor nodes is required.

The role of the MCU on the central node is to serialize sensor data from multiple branches of sensor nodes and transfer it to the Bluetooth module through a UART interface. The MCU accommodates a dedicated SPI module for each sensor branch (for our application, four SPI modules are used) to minimize processing time.

The Bluetooth module provides a standardized wireless link between the motion capture system and the processing device (this can be a smartphone, tablet, laptop or personal computer). Depending on power consumption constraints and application throughput, either classic or low-energy Bluetooth can be used.

An IMU is also integrated into the central node to efficiently use space, as it will be positioned on the body and used for body-movement measurements.

## 4. Sensor System Implementation in Clothing

The architecture described in Section 3 comprises multiple electronic components placed on the whole body for movement measurement, which need to be integrated into clothing without affecting wearability. The size and current consumption always have to be considered when designing wearable electronics. However, the main challenge of the sensor network implementation in the clothing addressed in this paper is the integration of wired connections.

For efficient power delivery, wire resistance must be kept at a minimum. Metallic wires provide excellent electrical conductivity. However, their incorporation in fabric poses multiple challenges for mechanical durability, flexibility and electrical stability during physical activities and cleaning.

In e-textile (electronics textile) projects for electrical connections, different kinds of conductive fibers, threads and fabrics can be used in combination with textile techniques such as sewing, weaving and knitting. Most of these materials are based on blending materials with good electrical properties and materials with good mechanical properties (flexibility and strength). Listings and short descriptions of different kinds of conductive e-textile materials and manufacturers can be found in [18].

Steel threads are made of fine fibers of an alloy of iron and carbon. Steel fibers, spun together, make mechanically strong, conductive and heat resistant threads, which can also be mixed with other fibers to vary the resistance. Compared to other metals, steel has a high resistance and, because of the heat resistance, it is often used for heating. Multiple threads can be used in parallel to increase the conductivity of the connection. For one of our prototypes, the stainless steel thread from Sparkfun was used. It features 28 Ω/ ft (92 Ω/ m) resistance and, to meet requirements for the resistance of power connections, 40 stainless steel threads had to be used. This considerably increases the weight of the wearable system and reduces the flexibility of the clothing. Therefore, alternative solutions were pursued.

To reduce the weight of the connections, materials with better conductivity are used. Copper has the second best conductivity of metals (59.6 MS/m)—almost as high as silver (63.0 MS/m)—but it has low mechanical durability compared to steel. To reduce mechanical stress on copper wires, they are combined with mechanically durable materials such as polyester ribbon [19]. These ribbons are elastic and mechanically durable, providing low-resistance metallic connections with a resistance of 0.4 Ω/ m.

To connect conductive materials to PCBs, multiple techniques, including soldering or sewing, are used. However, in these cases, electronics are permanently attached to the clothing. For removable electronics, specialized connectors for integration in the textile are used. Some examples are given in [20,21,22,23].

In this study, the approach with removable electronic components was implemented. Sensor nodes and the master hub, including the battery, were placed in rigid 3D-printed plastic housings, sewn onto the clothing according to the sensor placement described in Section 2.

## 5. Human Skeleton 3D Model and Movement Reconstruction

In this study, the human body approximation is given with a 3D stick figure model, as illustrated in Figure 6. This approximation is defined by multiple anatomical landmarks, indicating major joints of the human skeleton (marked with green circles in Figure 6), and corresponds to the visual marker positions used in this study for the optical sensor system to capture body movements.The movement of such a 3D model is fully defined by the time-varying coordinates of joints and, if the lengths and relations of skeleton rigid segments are known at each point in time, the 3D model can be reconstructed.

For the IMU sensor suit approximation model proposed in Figure 6, the installation is more convenient for the user, as it is not necessary to place the markers accurately, and each segment orientation is the same for the whole segment. However, compared to the coining joint marker coordinates, the 3D model reconstruction using only segment orientations is complex.

Steps for the proposed skeleton 3D model reconstruction are shown in Figure 7. They require prior knowledge of human body proportions and sensor orientations with respect to the patient’s body (used for the pose calibration). Measurements of the body and the pose calibration are acquired before performing physical activities, and usually do not change during the physical activity. As visual markers coincide with the joint locations in the 3D skeleton model (Figure 6), for convenience, the body measurements can be obtained with the digital cameras used in this study as a reference measurement system.

The reconstruction of body movements is based on continuous measurements of body segment orientations. A convenient representation for segment rotation is a unit quaternion
(1)$$q=\left[q_{w};q_{x};q_{y};q_{z}\right]=\left[\cos\left(\theta /2\right);n\cdot \sin\left(\theta /2\right)\right]$$
where θ is the rotation angle around axis n. Quaternions can be used to apply rotation to vectors v representing segments in the base model by using either the Hamilton product (quaternion multiplication) or a quaternion algebraic manipulation into a rotation matrix R [24]: (2)$$v^{\prime }=Rv$$
(3)$$R\left(q\right)=\left[\begin{array}{ccc} q_{w}^{2}+q_{x}^{2}-q_{y}^{2}-q_{z}^{2} & 2q_{x}q_{y}-2q_{w}q_{z} & 2q_{x}q_{z}+2q_{w}q_{y}\\ 2q_{x}q_{y}+2q_{w}q_{z} & q_{w}^{2}-q_{x}^{2}+q_{y}^{2}-q_{z}^{2} & 2q_{y}q_{z}-2q_{w}q_{x}\\ 2q_{x}q_{z}-2q_{w}q_{y} & 2q_{y}q_{z}+2q_{w}q_{x} & q_{w}^{2}-q_{x}^{2}-q_{y}^{2}+q_{z}^{2} \end{array}\right]$$

Quaternions representing body segment orientations during physical activities are acquired with the wearable IMU sensor system proposed in this article. IMU sensor locations based on considerations in Section 2 are shown in Figure 6, with red circles illustrating which orientations of the body segments can be obtained with the IMU system proposed in this paper. It is evident that visual markers and IMU positions in Figure 6 may differ. For example, the segment connecting the shoulder with the elbow differs, thereby introducing a mismatch between skeleton models. As a result, the 3D model reconstructed from IMU orientations is not identical to the 3D model reconstructed from the joint markers, but it contains the same amount of information.

Additionally, a base point has to be selected relative to which 3D Model coordinates of the skeleton are constructed and compared. The two most commonly used reference points are the neck and the sacrum.

## 6. Experimental Setup, Scenario and Results

### 6.1. Prototype of Wearable Sensor Clothing

The proposed IMU-based system prototype is made on top of a tight-fitting fabric tracksuit shown in Figure 8a and Figure 9). It accommodates BNO055 smart motion sensors, covering joint segments chosen in Section 2, and using the architecture of the system described in Section 3. The head sensor was not included in the prototype for the system assessment because it is not crucial for the physical activities selected in the net section. However, a dedicated connector for the head sensor is reserved on the master hub for future applications.

Sensor nodes are attached to the closing using 3D-printed plastic housings (outer dimensions: 25 × 18 × 7 mm), as shown in Figure 8b. Housings used for sensor placement are not humidity-resistant. However, they provide a convenient way of removing electronic components before washing and returning them when it is done.

The master node accommodates the BGM113 Blue Gecko Bluetooth module, MSP430F5438A MCU, BNO055 IMU and an 1100 mAh lithium-ion battery, all packed in a 3D-printed enclosure (outer dimensions: 55 × 55 mm), which is placed on the back of the body, near to the neck. Sensor nodes are organized in branches, as shown in the picture, to optimize wiring and data acquisition. Sensors are connected using elastic wires embedded in a ribbon from Ohmatex A/S. Wires are connected to nodes with right-angle 4-pin Micro Lock Plus connectors from Molex^®^. Four-wire connections (+3.3 V, SPI data, SPI clock, GND) are used for sensor data readout with a new solution of multi-branch enhanced SPI daisy chain topology and sensor powering.

BNO055 sensor nodes are configured to operate in nine degrees of freedom (NDOF) data fusion mode providing fused absolute orientation data from an accelerometer, a gyroscope and a magnetometer in quaternion form. In the NDOF mode, the fast sensor calibration is turned on, resulting in the quick calibration of all sensors and high output data accuracy.

Each sensor node is programmed to provide 11 bytes of data during each sampling event. The data are organized in the following order: quaternion (8 bytes), calibration status (1 byte), cyclic redundancy check (2 bytes). The master hub collects data from 15 sensor nodes and transfers them, using Bluetooth Low Energy protocol notifications, to the auxiliary processing device on which the 3D model is constructed.

The 3D model reconstruction, using orientation data from IMU sensors, requires a predefined base model mapping joint and segment relations at the base posture. In the experimental setup, the model illustrated in Figure 6 standing in straight posture and facing North was used.

IMU measurements of the wearable sensor clothing are in unit quaternion form. Quaternions $q_{sensor}$ provided by BNO055 sensors represent the orientation of the IMU local reference frame defined by sensor axis with respect to an absolute reference frame, defined by the direction of magnetic North and vertical direction of gravity. To obtain orientations of body parts $q_{body}$ with respect to the base model, a pose-compensating quaternion $q_{pose}$ is applied to the corresponding $q_{sensor}$, using quaternion multiplication ⨂
(4)$$q_{body}=q_{sensor}⊗q_{pose}^{-1}$$
where $q_{pose}^{-1}$ is the inverse quaternion of $q_{pose}$.

The $q_{pose}$ is obtained during a pose calibration procedure when a human wearing the IMU sensor clothing is standing in the base posture facing North. The orientations of the body parts in the North-facing base pose $q_{north}$ have to be predefined by the base model. Then, $q_{pose}$ can be calculated as follows
(5)$$q_{pose}=q_{sensor}⊗q_{north}^{-1}$$
where $q_{sensor}$ is sensor orientation during the pose calibration.

The accuracy of the pose calibration is limited by the accuracy of the posture, which is taken during the pose calibration procedure. Human posture deviations from the predefined base model result in pose reconstruction errors.

### 6.2. Experiments

The assessment of the performance of the system proposed in this article is based on the results obtained by testing the system itself and comparing it with the SMART DX motion detection system from BTS Bioengineering Corp. Four different body movement activities were used for tests: squat, lunge, push-up and bend. The chosen activities exploit multiple joint degrees of freedom with the highest possible amplitude. They were performed in sequential order, repeating each activity at least five times. The experimental studies were performed in cooperation with the Latvian Academy of Sport Education.

BTS SMART DX system provides high-accuracy (<0.1 mm) motion capture for motion analysis using digital cameras (up to 2048 × 2048 pixels) which record at up to 2000 frames per second and are equipped with powerful infrared illuminators to guarantee exceptional performance, even in adverse conditions. Seventeen markers for the BTS SMART DX system were placed on the person’s body according to the points defining the human 3D model in Figure 6, and recorded with 250 frames per second during physical activities. In parallel, a laptop running a Python program was used to calibrate, visualize and log incoming data from the proposed IMU sensor system. In these experiments, the configuration with fifteen BNO055 sensors placed on the body locations, also shown in Figure 6, was used. The test environment with the experimental setup is shown in Figure 9.

### 6.3. Results

From the performance evaluation of the system itself, the main bottleneck of the sensor data transmission is the UART connection node between the Bluetooth module and the MCU on the central. The application throughput of the Bluetooth Low Energy is up to 1.4 megabits per second [17] (for classical Bluetooth, it is even higher) and baudrates of the SPI range up to several megahertz. However, UART baudrates supported by microcontrollers are mostly lower. In the prototype of the wearable sensor clothing, the MSP430F5438A MCU with the maximum baudrate of 480,600 baud/s [25] was used. With the UART configuration for eight data sections, no parity bit, 1 start bit and 1 stop bit for the maximum data rate of the system is 48.060 kilobytes per second. This could provide 48-sensor sampling at 100 Hz or 15-sensor sampling at up to a 320.4 Hz frame rate if 10 bytes per sensor are transmitted. In this parameter, the proposed system outperforms other systems in Table 2.

The accuracy of the proposed system is limited by BNO055 MEMS sensor technology and embedded proprietary data calibration and fusion algorithms. In experimental tests, less than 1 degree accuracy for static orientations and less than 2 degree accuracy for dynamic orientations was observed in the range of 90 degrees, which complies with other system accuracies and with the experimental study in [26].

It should also be mentioned that the proposed solution is more energy-efficient and provides more than 3 h of operation with an 1100 mAh battery, as well as being cheaper than the systems discussed in Table 2. The estimated price is 500 EUR for a 15-sensor-node configuration.

For the assessment and comparison, a dataset with 15 IMU sensor orientations at 25 fps (frames per second) and 17 visual marker positions at 250 fps was obtained. The dataset contains marked data, obtained simultaneously with two motion-capture systems (sensor clothing presented in this paper and BTS Bioengineering SMART DX) during four physical activities (squat, lunge, push up and bend). The dataset is used to reconstruct the skeleton 3D model and evaluate the performance of the IMU sensor clothing described in this article.

The 3D model of the human body (defined in Figure 6) is the same for both systems. For the BTS SMART DX camera system, it is defined by joint coordinates (green circles). For the IMU sensor system, it is defined by lengths and orientations of segments defined by joint connections (black lines). At the beginning of the experiment, human body proportions were measured by calculating the distance between the joint markers captured by the BTS SMART DX system. These measurements are used in a body-pose reconstruction for the IMU sensor system to ensure that both models have the same properties, and a snapshot of the pose reconstruction results from both systems is given in Figure 10.

According to sensor placement in Figure 6, some segment orientations defined by joint coordinates acquired with the camera system differ from corresponding body part orientations acquired with IMU sensor clothing. Therefore, a direct comparison of pose coordinates is not applicable. However, some specific parts of the skeleton and can be compared—for example, the knee joint angle defined by connecting segments (7-2)-(2-1) for the left leg and (8-5)-(5-4) for the right leg. Knee angles θ of both 3D skeleton models are calculated with dot product of upper and lower leg segments
(6)$$\theta =\arccos\frac{v_{\mathrm{upper}}\cdot v_{\mathrm{lower}}}{\left|\right|v_{\mathrm{upper}}\left|\right|\cdot \left|\right|v_{\mathrm{lower}}\left|\right|},$$
where $v_{\mathrm{upper}}$ and $v_{\mathrm{lower}}$ are vectors representing upper and lower leg segments, respectively. The results of the knee angle comparison are shown in Figure 11. The mean error between both systems for the knee angle during squats was estimated at 1.89 degrees.

In addition, we would like to note that, during the post-processing phase of the BTS SMART DX motion detection system, the loss of multiple visual markers caused by occlusion was observed. The overview of defective frames (if at least one marker is not detected in the frame) and markers lost (number of lost markers from all markers in all frames) is given in Table 3. For the IMU sensor clothing measurements, data loss was not observed.

## 7. Discussion and Conclusions

This work dealt with the problems of the development and validation of a low-cost, user-friendly wearable sensor system for body-movement measurements. A system based on the architecture described in Section 3, with IMU sensors organized in multiple branches, was developed. The sensor placement was selected, taking the anatomical properties of the human body and physical exercises chosen for experiments mentioned above into account.

The main result of the paper is an efficient hardware and middleware solution for IMU-based sensor clothing. The IMU sensor clothing obtains orientations of corresponding human body segments, which can be used for an application-specific patient’s body movement analysis carried out by scientists and doctors. In the future, research on the assessment of the proposed wearable-sensor network for patients’ full-body activity recognition and motion analysis using machine learning and artificial intelligence is planned. Current studies indicate that this has potential for telemedical assessment for health risks and rehabilitation [15,27].

In this study, the concept of the wearable sensor is implemented with low power, lightweight electronic components and stretchable wired connections between nodes. The described prototype is powered by a single lithium-ion battery, in which all nodes are removable before washing.

Compared to the commercial IMU motion-tracking suits listed in Table 2, the results of the study showed that the performance of the prototype is competitive by providing a sampling of up to 48 sensors with 100 fps, 1 degree of static and 2 degrees of dynamic accuracy, and more than 3 h of operation with 1100 mAh battery. In addition, the estimated price of the 15-sensor configuration (500 EUR) is many times lower than similar commercial IMU motion-tracking suits.

In the future, humidity protection and sensor clothing washing are important issues. The current prototype does not facilitate any humidity protection, and all electronic components must be removed before washing. However, we imagine that sealed connectors would considerably increase the size and the weight of the system, reducing the wearability. Therefore, an alternative solution would be to completely seal all electronic components (including wires) in a thin layer of waterproof and stretchable polymer, for example, thermoplastic polyurethane (TPU).

The concept was successfully evaluated by testing it over four different physical activities. The results confirm that the proposed wearable clothing could be used as a relatively low-cost alternative to measure body motion and reconstruct a 3D model of a skeleton. We note that the head sensor was not used in this study, as the head movement was not crucial for the physical activities used for the assessment of the proposed sensor system. However, for future usage, the addition of the head sensor is supported both by the system architecture described in Section 3 and the prototype developed in Section 6.1.

In the Section Results, the differences between segment orientations captured by the IMU sensor system and digital camera system were noted. In the future, to avoid a mismatch of human 3D body measurements and exploit the accuracy of digital camera systems for the assessment of IMU sensor systems, visual marker positions should be chosen more specifically to obtain precise segment orientations measured by the IMU sensor system.

Unfortunately, the COVID-19 pandemic affected the ability to collect a wider set of data, especially by involving patients in healthcare institutions, which would have allowed for a more comprehensive evaluation and a comparison of the proposed solution with other alternatives.

The proposed solution outperforms similar IMU-based systems that are currently available in terms of price and characteristic parameters, and, in comparison with the BTS SMART DX system, no data loss is observed. During some specific exercises, the optical sensor system lost individual markers in up to two-thirds of the frames. It is undesirable in medical applications. Additionally, compared to other IMU sensor systems using wires for communication, our solution facilitates a lightweight solution for permanently embedding elastic wired connections in the clothing, improving the wearability of the sensor system.

In the future, taking into account that the used architecture easily allows the addition of other types of sensors, the proposed wearable sensor clothing is planned to be supplemented with EMG sensors, similar to that used in [28,29,30].

## Figures and Tables

**Figure 1 sensors-21-02068-f001:**
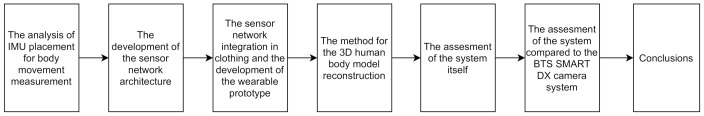
Flowchart of the proposed research.

**Figure 2 sensors-21-02068-f002:**
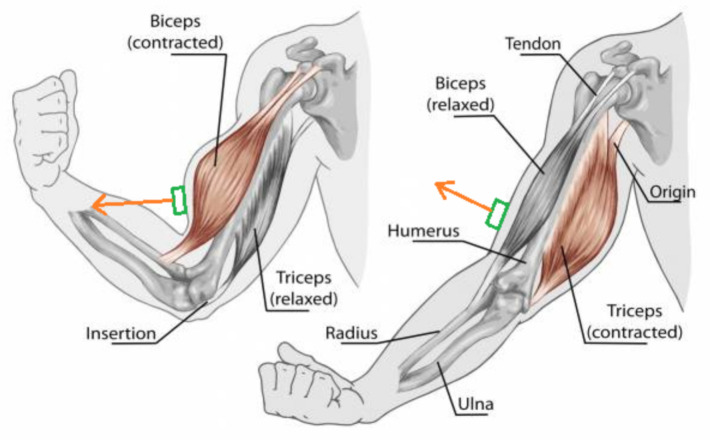
Sensor placement error.

**Figure 3 sensors-21-02068-f003:**
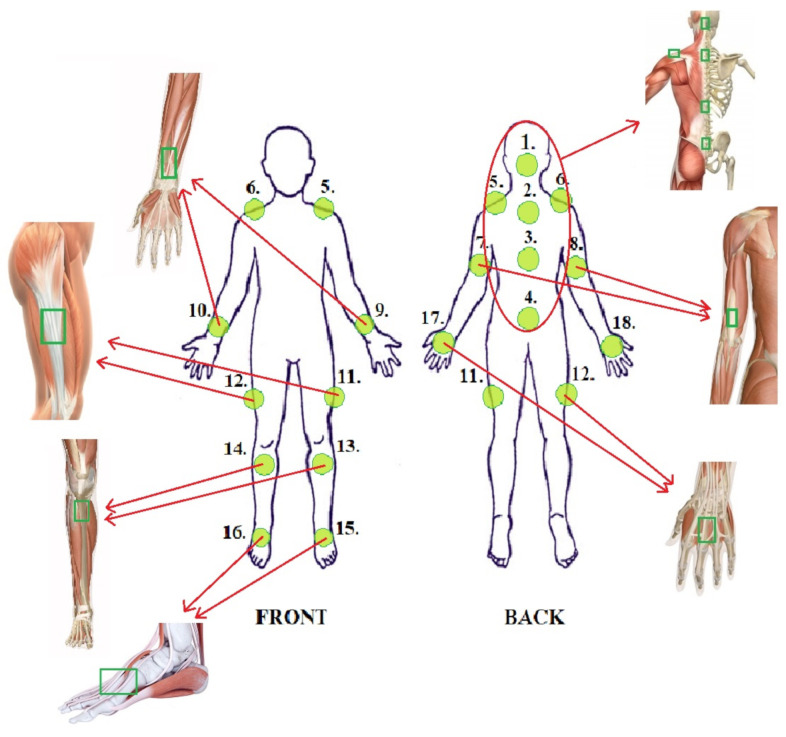
Sensor placement.

**Figure 4 sensors-21-02068-f004:**
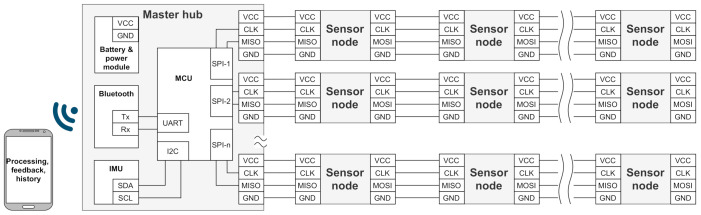
The architecture of sensor network for motion tracking system.

**Figure 5 sensors-21-02068-f005:**
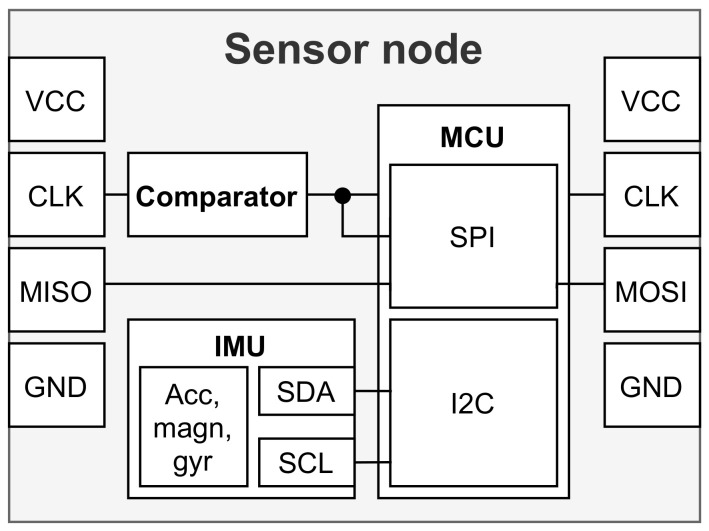
Schematic of a single IMU snsor node.

**Figure 6 sensors-21-02068-f006:**
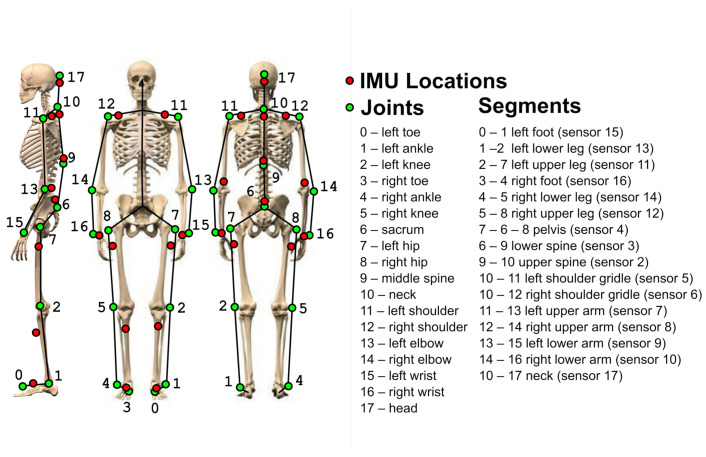
Human body approximation model.

**Figure 7 sensors-21-02068-f007:**
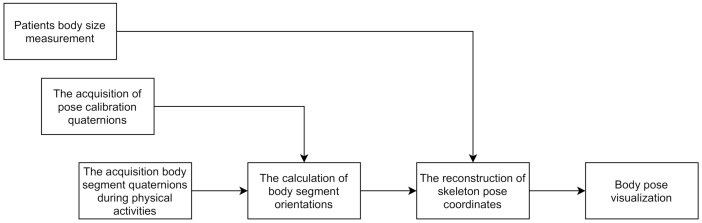
Flowchart of the proposed method for human skeleton 3D model and movement reconstruction

**Figure 8 sensors-21-02068-f008:**
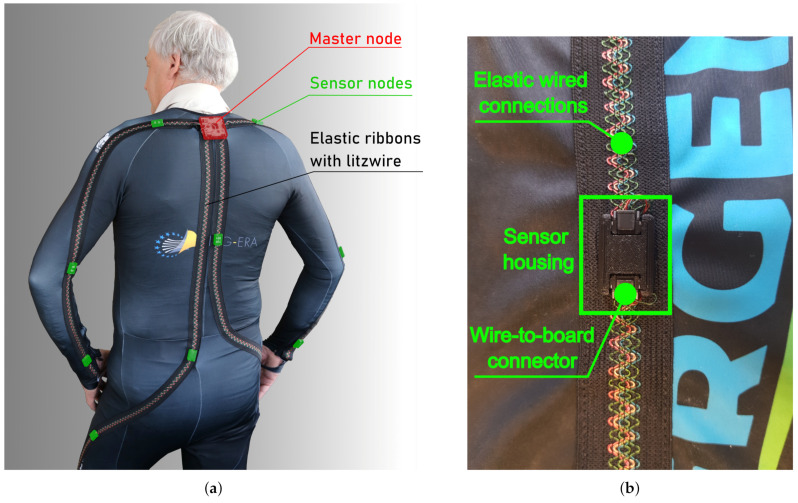
Experimental setup. (**a**) The sensor clothing with multi-branch IMU sensor network. (**b**) The custom 3D-printed housing used for sensor placement.

**Figure 9 sensors-21-02068-f009:**
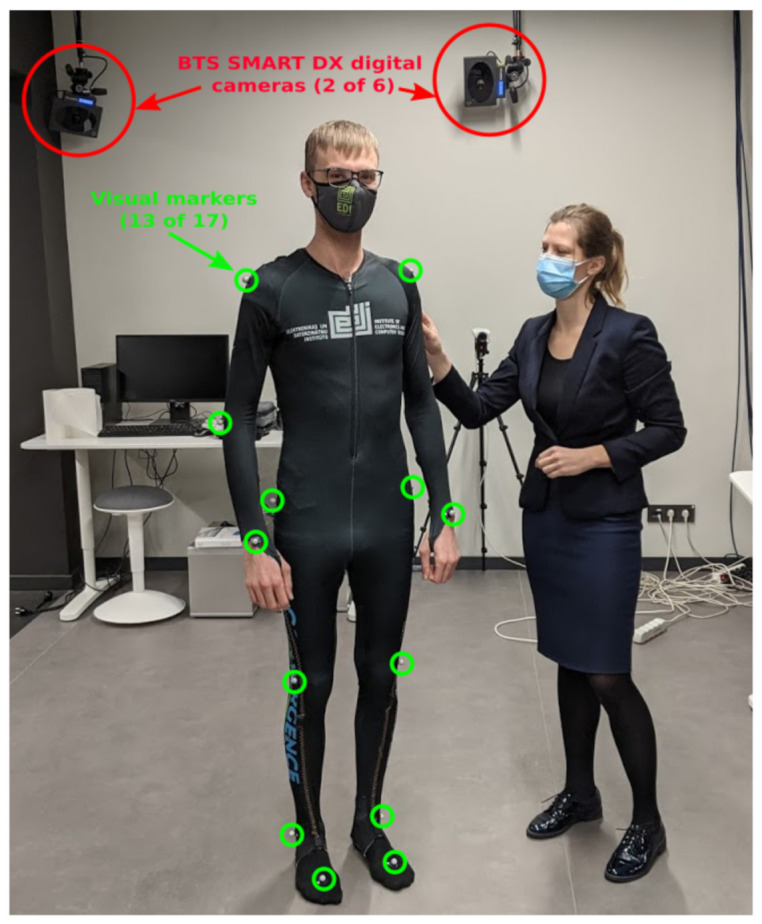
The sensor clothing testing environment with BTS SMART DX system.

**Figure 10 sensors-21-02068-f010:**
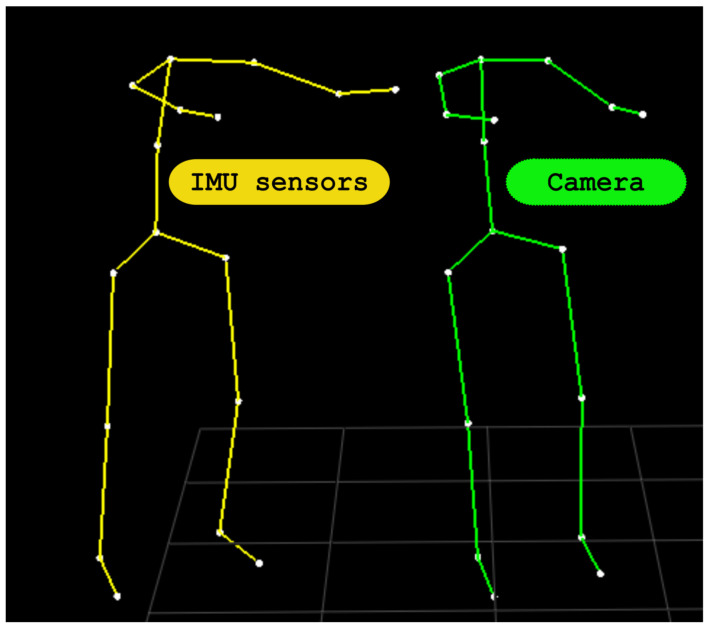
Human body models reconstructed from IMU sensor clothing (left) and BTS SMART DX camera system (right). Sacrum used for the base point.

**Figure 11 sensors-21-02068-f011:**
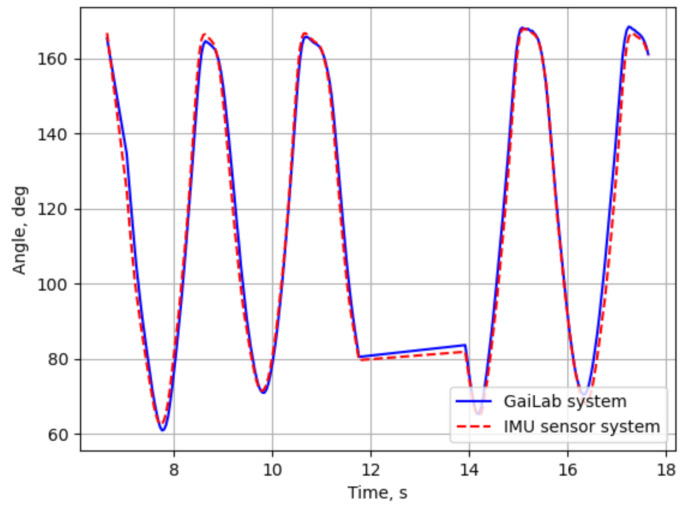
Comparison of knee angles during squats.

**Table 1 sensors-21-02068-t001:** Summary of motion-tracking technologies.

Technologies	Accuracy	Refresh Rate	Constraints
Acoustic	Low	Varies with speed of sound	Line of sight, acoustic interference
Electromagnetic	High	Up to few hundreds of Hz	Working volume, metal object interference
Inertial	High	Up to few hundreds of Hz	Magnetic object interference
Mechanical	High	Up to few hundreds of Hz	Mechanical arm paradigm, working volume
Optical	High	Up to few tens of Hz	Line of sight, infrared and visible light interference

**Table 2 sensors-21-02068-t002:** Comercial inertial motion unit (IMU) motion-tracking suits (P-Pitch, R-Roll, Y-Yaw).

Brand Title	Version	Wired/Wireless	Dynamic Accuracy	Static Accuracy	Sensors	Hardware Cost
Xsens	Lycra suit (Link)	Wired	P/R/Y: 1°	P/R: 0.2°, Y: 0.5°	17	9225$
Xsens	Strap-based (Awinda)	Wireless	P/R/Y: 1°	P/R: 0.2°, Y: 0.5°	17	8180$
Shadow Motion	Shadow motion capture system	Wired	P/R/Y: 2°	P/R/Y: 0.5°	17	4000$
STT Systems	iSen system	Wireless	P/R/Y: <2°	P:<0.5°R/Y: <2°	16	10,584$
Nansense	INDIE full-body motion capture suit	Wired	P/R: 0.7°,Y:1.4°	P/R: 0.5°,Y:1°	16	6300$
Rokoko	Smartsuit Pro	Wired	P/R/Y: 1.5°	Not measured	19	2495$
Perception Neuron	Perception Neuron Pro	Wireless	Not measured	P/R: 1°, Y: 2°	17	4000$
AiQ-Synertia	IGS Cobra Suit	Wired or wireless	Not measured	P/R: 1°, Y: 2°	22	14,450$

**Table 3 sensors-21-02068-t003:** Overview of defective frames of camera system caused by loosing markers during multiple physical activities.

Activity	Total Count of Frames	Defective Frames	Total Count of Markers	Markers Lost
Lunges	6016	4058 (67.45%)	102,272	14,982 (14.65%)
Bends	6766	669 (9.89%)	115,022	2142 (1.86%)
Squats	5987	877 (14.65%)	101,779	3501 (3.44%)
Push ups	6521	4695 (72.0%)	110,857	17,098 (15.42%)

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
