# Peer review of "Wearable Sensor Clothing for Body Movement Measurement during Physical Activities in Healthcare"

_sensors, 2021, doi:10.3390/s21062068_

Round 1

Reviewer 1 Report

-Discussion Section is missing;;; It is too short;;;;
-please add block diagram of the proposed research step by step ;;; what is the result of paper?;;;
-please add block diagram of the proposed method;;;
-please add sentences about future analysis;;;
-references should be 2018-2021 Web of Science about 50% or more  (30 references at least);;
-Please compare with other methods, justify. Advantages or Disadvantages;;;
for example:

1) Pattern Recognition of Single-Channel sEMG Signal Using PCA and ANN Method to Classify Nine Hand Movements, 
SYMMETRY-BASEL, Volume: 12 Issue: 4, Article Number: 541, Published: APR 2020, DOI: 10.3390/sym12040541

2) Recognition of Sedentary Behavior by Machine Learning Analysis of Wearable Sensors during Activities of Daily Living for Telemedical Assessment of Cardiovascular 
Risk, Sensors 2018, 18, 3219; DOI: 10.3390/s18103219

-Conclusion: point out what are you done;;;;

Author Response

Dear Reviewer,

We are grateful for the attention you paid to our manuscript and we really appreciate the positive comments with the valuable suggestions that helped us to improve our manuscript.

Below we include the point-by-point response to your observations. The changes done in the manuscript are tracked in the main text.

Reviewer comment (RC): Discussion Section is missing. It is too short.

Our response (OR): We acknowledge the Reviewer for the feedback: we have expanded the discussion part in Section 7 “Discussion and Conclusions”.

RC: Please add block diagram of the proposed research step by step.

OR: The step-by-step block diagram of the proposed research is included as Fig. 1 in Section 1 “Introduction”.

RC: What is the result of paper?

OR: Considering your question Section 7 “Discussion and Conclusions” was modified stating the main result of the paper clearly in the line 365.

RC: Please add block diagram of the proposed method!

OR: We acknowledge the Reviewer for the suggestion: a block diagram of the proposed method for 3D human body model reconstruction from IMU data is included as Fig. 7 in Section 5 “Human skeleton 3D Model and movement reconstruction”.

RC: Please add sentences about future analysis!

OR: We appreciate the Reviewers suggestion. We have added a paragraph of future analysis in Section 7 “Discussion and Conclusions” lines 377-382 and 385-389.

RC: References should be 2018-2021 Web of Science about 50% or more (30 references at least)

OR: Thank you for noting this suggestion. The reference list has been reviewed and updated with more topical research. 16 of 30 references are 2018-2021 Web of Science.

RC: Please compare with other methods, justify. Advantages or Disadvantages. For example:

1) Pattern Recognition of Single-Channel sEMG Signal Using PCA and ANN Method to Classify Nine Hand Movements, SYMMETRY-BASEL, Volume: 12 Issue: 4, Article Number: 541, Published: APR 2020, DOI: 10.3390/sym12040541

2) Recognition of Sedentary Behavior by Machine Learning Analysis of Wearable Sensors during Activities of Daily Living for Telemedical Assessment of Cardiovascular Risk, Sensors 2018, 18, 3219; DOI: 10.3390/s18103219

OR: Thank you for pointing out this research about wearable sensors. We acknowledge the Reviewers suggestion. A discussion of different sensor network architectures (including the bus architecture described in the 2nd reference) is added to Section 3 “Sensor network architecture” starting line 115. Also, a comparison of the proposed system with similar commercial IMU sensor suits was added to Section 7 “Discussion and Conclusions” starting line 372.

Considering the 1st reference, we would like to note that our study is about the development and the assessment of a wearable sensor system for body movement measurement, and the usage of the proposed system for automated pose recognition is currently planned for future studies.  

RC: Conclusion: point out what are you done!

OR: Thank you for noting this observation. Section 7 “Discussion and Conclusions” is expanded to state more clearly the results of the research starting line 390.

We think to have addressed all Reviewers’ comments, clarifying the doubtful aspects in the manuscript. We thus believe that the manuscript is suitable for publication.

Reviewer 2 Report

  1. In section 3, you have discussed about the power supply to your design. I could not find any detailed information about the power supply. The title of the paper is the wearable sensor. How have you implemented this concept in clothing, the detail is required?
  2. How could you preserve your connection of the conductive wire to PCB's from humidity? Is your design water resistant? If not how do you manage to overcome these issues?
  3. Have you put any nodes to monitor the movement of the head?

Author Response

Dear Reviewer,

We are grateful for the attention you paid to our manuscript and we really appreciate the positive comments with the valuable suggestions that helped us to improve our manuscript.

Below we include the point-by-point response to your observations. The changes done in the manuscript are tracked in the main text.

Reviewer comment (RC): In section 3, you have discussed about the power supply to your design. I could not find any detailed information about the power supply.

Our response (OR): Thank you for this remark. Additional details about power management and power supply are included in Section 3 "Sensor network architecture" starting line 153 and Subsection 6.1 “Prototype of wearable sensor clothing” line 262.

RC: The title of the paper is the wearable sensor. How have you implemented this concept in clothing, the detail is required?

OR: We acknowledge the Reviewers suggestion: Section 4 “Electrical connections solution for wearable applications” was renamed to “Sensor system implementation in clothing” in line 170 and general considerations about the implementation of the wearable concept were included in this section starting lines 171 and 202. Detailed information regarding the sensor implementation in the clothing for the prototype is included in subsection 6.1 “Prototype of wearable sensor clothing” starting line 257 and in Fig 8b.

RC: How could you preserve your connection of the conductive wire to PCB's from humidity? Is your design water resistant? If not how do you manage to overcome these issues?

OR: Thank you for noting this important question for wearable electronics. The current prototype does not facilitate any humidity protection. Before washing all sensor nodes and the master hub including the battery must be removed. However, for future improvements to overcome the issue of humidity protection we have added a brief discussion in Section 7 “Discussion and Conclusions” starting line 377.

RC: Have you put any nodes to monitor the movement of the head?

OR: Yes, thank you for noting this! The prototype developed in this study facilitates a dedicated connector for the head sensor. However, it was not used in the assessment of the system, because in this study the head movement was not considered crucial for selected physical activities. Considering your question, a remark about the head sensor is added in Subsection 6.1. starting line 253 and a discussion for the future was added in Section 7 starting line 385.

We think to have addressed all Reviewers’ comments, clarifying the doubtful aspects in the manuscript. We thus believe that the manuscript is suitable for publication.

Reviewer 3 Report

The paper deals with the design of a wearable wireless system for measuring human body activities. It is well written, the description of the design is enough detailed and supported by adequate measurement results.

I suggest to accept it for the publication on Sensors

Author Response

Dear Reviewer,

We are grateful for the attention you paid to our manuscript and we really appreciate the positive comments!

Reviewer 4 Report

BRIEF SUMMARY

The study describes the development of a wearable wireless system for measuring human body activities, consisting of small inertial sensor nodes and the main hub for data transmission via Bluetooth. I congratulate authors on their work. This is a well-written paper with informative figures and tables. The paper contributes significantly to the increasing trend of measuring human movement outside laboratory. Overall, I found the topic timely and clinically important. Only small suggestions are listed below

SPECIFIC COMMENTS

  1. It is quite difficult to figure out what is aim of the study. Please expand/state clearly more on this point in the introduction.
  2. Better description is needed how you compare two systems. What metrics you compare etc.?
  3. Discussion is limited in scope. Please add more information about study results and how your system compares with other similar systems available on the market.

Author Response

Dear Reviewer,

We are grateful for the attention you paid to our manuscript and we really appreciate the positive comments with the valuable suggestions that helped us to improve our manuscript.

Below we include the point-by-point response to your observations. The changes done in the manuscript are tracked in the main text.

Reviewer comment (RC): It is quite difficult to figure out what is aim of the study. Please expand/state clearly more on this point in the introduction.

Our Response (OR): Thank you for this feedback. We expanded the aim statement in the introduction starting line 60 to clarify the aim of the study.

RC: Better description is needed how you compare two systems. What metrics you compare etc.?

OR: Thank you for noting this remark. Additional description of the metrics how two systems are compared was added in Subsection 6.3 “Results” starting line 349.

RC: Discussion is limited in scope. Please add more information about study results and how your system compares with other similar systems available on the market.

OR: We acknowledge the Reviewer for the suggestion. Based on specific parameters described in Subsection 6.3 “Results”, a comparison of the proposed system and commercial IMU motion tracking suits was included in Section 7 “Discussion and Conclusions” starting line 372.

We think to have addressed all Reviewers’ comments, clarifying the doubtful aspects in the manuscript. We thus believe that the manuscript is suitable for publication.

Round 2

Reviewer 1 Report

-please compare your approach with

Grasp Posture Control of Wearable Extra Robotic Fingers with Flex Sensors Based on Neural Network
By:Setiawan, JD (Setiawan, Joga Dharma)[ 1,2 ] ; Ariyanto, M (Ariyanto, Mochammad)[ 1,2 ] ; Munadi, M (Munadi, M.)[ 1 ] ; Mutoha, M (Mutoha, Muhammad)[ 1 ] ; Glowacz, A 
(Glowacz, Adam)[ 3 ] ; Caesarendra, W (Caesarendra, Wahyu)[ 1,4 ]
ELECTRONICS
Volume: 9 Issue: 6
Article Number: 905
DOI: 10.3390/electronics9060905
Published: JUN 2020
Document Type:Article

or

Recognition of Sedentary Behavior by Machine Learning Analysis of Wearable Sensors during Activities of Daily Living for Telemedical Assessment of Cardiovascular Risk
By:Kantoch, E (Kantoch, Eliasz)[ 1 ]
SENSORS
Volume: 18 Issue: 10
Article Number: 3219
DOI: 10.3390/s18103219
Published: OCT 2018

Author Response

Dear Reviewer,

Thank you for the time paid for our work and valuable suggestions!

We have read both publications and in our opinion, the aim, contribution and results of the first publication are rather different from the essence of our article. Our results regarding the body pose measurement or reconstruction do not provide comparable information.

Regarding the second article, we see that the approaches to architecture are comparable and we have discussed this in our article in Section 3 starting line 130 in the main text. In order to be able to compare the results, we plan to perform new research using the wearable sensor network proposed in the current article involving machine learning analysis of wearable sensors during activities of daily living (discussed in the main article starting line 358). On the basis of these new results, we plan to prepare a new publication, and in this new publication, we will definitely consider the possibility to compare the results with the results of your mentioned publication.

Also, based on your review we have improved the English language and style throughout the article. 

We think to have addressed all Reviewers’ comments, clarifying the doubtful aspects in the manuscript. We thus believe that the manuscript is suitable for publication.

Reviewer 4 Report

Thank you.

Author Response

Dear Reviewer,

We are grateful for the attention you paid to our manuscript and suggestion for improving it.

Based on your suggestion for minor spellcheck, we have improved the English language and style throughout the article. 

We think to have addressed all Reviewers’ comments, clarifying the doubtful aspects in the manuscript. We thus believe that the manuscript is suitable for publication.